

# Sex differences in the response to angiotensin II receptor blockade in a rat model of eccentric cardiac hypertrophy

Élisabeth Walsh-Wilkinson, Marie-Claude Drolet, Charlie Le Houillier, Ève-Marie Roy, Marie Arsenault and Jacques Couet

Université Laval, Groupe de recherche sur les valvulopathies, Centre de recherche, Institut universitaire de cardiologie et de pneumologie de Quebec, Québec, Québec, Canada

## ABSTRACT

**Background.** Men and women differ in their susceptibility to cardiovascular disease, though the underlying mechanism has remained elusive. Heart disease symptoms, evolution and response to treatment are often sex-specific. This has been studied in animal models of hypertension or myocardial infarction in the past but has received less attention in the context of heart valve regurgitation. The aim of the study was to evaluate the development of cardiac hypertrophy (CH) in response to left ventricle (LV) volume overload (VO) caused by chronic aortic valve regurgitation (AR) in male and female rats treated or not with angiotensin II receptor blocker (ARB), valsartan. We studied eight groups of Wistar rats: male or female, AR or sham-operated (sham) and treated or not with valsartan (30 mg/kg/day) for 9 weeks starting one week before AR surgical induction.

**Results.** As expected, VO from AR resulted for both male and female rats in significant LV dilation (39% vs. 40% end-diastolic LV diameter increase, respectively; $p < 0.0001$) and CH (53% vs. 64% heart weight increase, respectively; $p < 0.0001$) compared to sham. Sex differences were observed in LV wall thickening in response to VO. In untreated AR males, relative LV wall thickness (a ratio of wall thickness to end-diastolic diameter) was reduced compared to sham, whereas this ratio in females remained unchanged. ARB treatment did not prevent LV dilation in both male and female animals but reversed LV wall thickening in females. Systolic and diastolic functions in AR animals were altered similarly for both sexes. ARB treatment did not improve systolic function but helped normalizing diastolic parameters such as left atrial mass and E wave slope in female AR rats. Increased LV gene expression of *Anp* and *Bnp* was normalized by ARB treatment in AR females but not in males. Other hypertrophy gene markers (*Fos, Trpc6, Klf15, Myh6* and *Myh7*) were not modulated by ARB treatment. The same was true for genes related to LV extracellular matrix remodeling (*Col1a1, Col3a1, Fn1, Mmp2, Timp1* and *Lox*). In summary, ARB treatment of rats with severe AR blocked the female-specific hypertrophic response characterized by LV chamber wall thickening. LV dilation, on the other hand, was not significantly decreased by ARB treatment. This also indicates that activation of the angiotensin II receptor is probably more involved in the early steps of LV remodeling caused by AR in females than in males.

**Subjects** Anatomy and Physiology, Cardiology

Corresponding author
Jacques Couet,
jacques.couet@med.ulaval.ca

**Keywords** Cardiac hypertrophy, Sex differences, Angiotensin receptor blockade, Aortic valve regurgitation, Rat, Volume overload, Echocardiography, Left ventricle

# INTRODUCTION

Heart diseases are among the leading causes of mortality for both men and women (*Roth et al., 2017*). Cardiac hypertrophy (CH) is an adaptive response to overload (pressure (PO) or volume (VO)) (*Swynghedauw, 1999*) and is an independent cause of morbidity and mortality from heart diseases. Heart failure (HF) in women occurs as frequently as in men but later in life and less from ischemic causes. Women are more likely to develop HFpEF, which is associated with diastolic dysfunction (impaired myocardial relaxation during filling) and concentric LV (left ventricle) remodeling (*Maric-Bilkan et al., 2016*). In preclinical models of pathological CH and/or HF such as in mice with transverse aortic constriction (TAC; LV PO model), males develop concentric LV hypertrophy (LVH) sooner than females. Myocardial remodeling in males then evolves more rapidly towards eccentric LVH and HFrEF than in females (*Regitz-Zagrosek et al., 2010*). A sexual dimorphism is thus present in the hypertrophic response to an overload in both patients and in pre-clinical models (*Blenck et al., 2016*). Sexually dimorphic response to PO (hypertension) and effects of treatment has been relatively well-documented. This is not the case for VO.

LV VO occurs when either the aortic or the mitral valve is regurgitating. Causes for regurgitation are multiple but worldwide, they are most frequently complications of acute rheumatic fever. Rheumatic valve diseases causing aortic regurgitation (AR) are still occurring at an alarming rate in the third-world and low/middle-income countries. In the Western world, they are still prevalent in poor, remote, Native American communities and in immigrants from countries where rheumatic fever is still endemic. The estimated burden worldwide of rheumatic valve diseases is estimated to more than 15 million existing cases with 280k new cases each year and 230k deaths (*Marijon et al., 2012*). Secondary moderate to severe AR also occurs in a small proportion of patients (5–10%) undergoing transcatheter aortic valve replacement (TAVR) (*Leon et al., 2016*). Since TAVR is now a procedure routinely performed, management of secondary AR is a developing concern.

LV remodeling in response to significant VO from experimental severe AR in male Wistar rats results in important eccentric hypertrophy (dilation) to accommodate the excess regurgitating aortic blood to pump (*Arsenault et al., 2002*; *Plante et al., 2003*). We recently observed in a rat model of chronic (6 months) LV VO caused by severe aortic valve regurgitation (AR), that female animals developed as much if not more CH than males (*Beaumont et al., 2017*). However, male LVs showed more dilation and worse contractile function than those of females. Interestingly, LV remodeling in AR female rats is characterized by a more important increase in LV wall thickness than in males (*Beaumont et al., 2017*). In another rat VO model (aorto-caval fistula), a faster progression toward HF was observed in males and resulted in poorer survival (*Dent, Tappia & Dhalla, 2010a*). At the cellular and molecular levels, we observed that male AR rat LVs showed an important down-regulation of many fatty acid oxidation genes and an up-regulation of glucose metabolism genes, whereas this characteristic energy metabolism switch did not happen in

females (*Beaumont et al., 2017*). Since sex steroids have a potent effect on differentiation, they could explain a large part of the sex dimorphism as observed in CH (*Leinwand, 2003*).

Activation of the tissue renin-angiotensin-aldosterone system (RAAS) system is a characteristic feature of the myocardial response to a pathological and chronic stress such as a significant valve regurgitation (VO) or a LV pressure overload such as in hypertension or aortic valve stenosis. We previously showed that blocking the RAAS in male AR rats could reduce development of LV hypertrophy (LVH), improve myocardial function and survival (*Plante et al., 2009*; *Plante et al., 2004a*; *Arsenault et al., 2013*). However, we did not investigate the benefits of inhibiting the RAAS in female AR rats. Here, we wanted to compare the hypertrophic response to treatment targeting the RAAS of animals of both sexes with a severe LV volume overload. We studied the effects of an angiotensin II receptor antagonist, valsartan, on the hypertrophic response to severe LV volume overload from AR in rats of both sexes and over a relatively short duration of two months in order to better differentiate early cardiac remodeling events between males and females with AR. We started treatment one week before AR induction instead as two weeks after as described in the chronic studies above in order to hopefully inhibit early features of LV remodeling under the control of the RAAS in our animals.

Our results suggest that angiotensin receptor blockade (ARB) with valsartan during the development of hypertrophy in female rats with severe AR partly abrogates LV wall thickening leading to a more eccentric remodeling similar to the one observed in males.

## METHODS

### Animals

Severe AR was induced in males (300–325 g) and females (200–225 g) Wistar rats (9–10 weeks of age) by retrograde puncture one or two aortic valve leaflets under echocardiographic guidance as previously described (*Arsenault et al., 2002*). Only animals with 50% and more regurgitation were included in the study. The regurgitant fraction was estimated by the ratio of the forward systolic flow time–velocity integral (VTI) to the reversed diastolic flow VTI measured by pulsed Doppler in the thoracic descending aorta. Eleven male and ten female Wistar AR rats received daily valsartan (30 mg/kg/d) mixed in unsalted peanut butter (1:50;w:w). Untreated animals received equivalent amount of peanut butter (5–6 animals/gr.). Treatment was started one week before AR induction. We made sure that peanut butter was consumed by all animals, daily. In addition, 24 sham-operated male and female Wistar rats (Sham or Sh) were used as controls and received treatment following the same regimen as AR rats. The protocol was approved by the Université Laval's Animal Protection Committee and followed the recommendations of the Canadian Council on Laboratory Animal Care.

### Echocardiography

An echocardiographic exam was performed two weeks after surgery to confirm AR severity (estimation of regurgitation and LV dimensions) and at the end of the protocol 8 week later as previously described (*Arsenault et al., 2013*; *Arsenault et al., 2002*; *Plante et al., 2003*). LV

mass estimated by echocardiography was calculated using the following equation.

$$1.04x((EDD + PW + SW)^3 - EDD^3)$$

EDD, PW, and SW are end-diastolic diameter, posterior wall thickness, and septal wall thickness, respectively. At the end of the protocol, the heart and the lungs were harvested and weighed. Heart chambers were dissected, weighted and the LV was then quickly frozen in liquid nitrogen and kept at −80 C until further use.

## Gene Expression Analysis by quantative RT-PCR

Total RNA was extracted using Trizol reagent as described elsewhere (*Champetier et al., 2009*). LV RNA samples were diluted to 500 ng/microliter. One microliter RNA (500 ng) was converted to cDNA using the QuantiTect Reverse Transcription kit (Qiagen). The cDNA obtained was further diluted 11-fold with water prior to amplification (final concentration corresponding to 4.54 ng/microliter of initial RNA). Five microliter diluted cDNA were amplified in duplicate by quantitative PCR in a Rotor-GeneTM thermal cycler (Corbett Life Science, Sydney, Australia). Pre-optimized primers were from QuantiTect (Qiagen) and IDT (Coralville, Iowa) (Table 1) and SsoAdvanced Universal SYBR Green Supermix (Bio Rad, Hercules, CA, USA) was used. Each run included one tube with water only (no template control) and a series of three 10-fold dilutions of a representative cDNA sample to check efficiency of the amplification reactions. We studied 5–6 animals/group. The six animals/group studied from the AR valsartan-treated groups were chosen randomly. Cyclophilin A gene (*Ppia*) was the housekeeping gene.

## Statistical analysis

Results are presented as the mean and the standard error of the mean (SEM). Statistical analyses were performed on the log of the data. Two-way ANOVA analysis was performed and Holm-Sidak's pos *t*-test was used for comparison between the groups (GraphPad Prism 8.02; GraphPad Software, Inc., San Diego, CA, USA). A Student's *t*-test was used when only two groups were compared. A *p*-value lower than 0.05 was considered significant.

# RESULTS

## Animal characteristics

Treatment with the angiotensin receptor blocker (ARB), valsartan, was initiated a week before surgery and lasted up until the end of the protocol 9 weeks later. Surgery itself, had no effects on body weight gain during the protocol. Valsartan treatment had no significant effects on heart total weight in sham-operated animals, males and females (Tables 2 and 3). The only significant difference was for the left atrial weight which was decreased by the ARB treatment in females.

Comparing AR rats to sham ones, every parameter measured with the exception of body weight and tibial length were significantly increased as summarized in Tables 2 and 3. Moreover, heart total and indexed weights were significantly reduced in female animals treated with the ARB (Table 3). Only a trend for a decrease was present in males. This was also true for the left and right ventricles, which were smaller in female AR rats treated

**Table 1** **Name and symbol of all primer pairs used for gene expression analysis by quantitative RT-PCR.** The table also includes catalogue numbers (from IDT or Qiagen) and the size of the amplicon.

| mRNA | Symbol | Catalog no. | Amplicon (bp) |
|---|---|---|---|
| procollagen-1 alpha-1 | Col1 | Rn.PT.58.7562513 | 134 |
| procollagen-3 alpha-1 | Col3 | Rn.PT.58.11138874 | 100 |
| fibronectin 1 | Fn1 | Rn.PT.58.18226984 | 114 |
| osteosarcoma viral oncogene homolog | Fos | QT01576330 | 73 |
| krüppel-like factor 15 | Klf15 | Rn.PT.58.12431283 | 129 |
| lysyl oxidase | Lox | Rn.PT.58.10677971 | 150 |
| matrix metalloproteinase-2 | Mmp2 | Rn.PT.58.44737355 | 87 |
| myosin, heavy polypeptide 6, cardiac | Myh6 | Rn.PT.58.8646063 | 150 |
| myosin, heavy polypeptide 7, cardiac | Myh7 | Rn.PT.58.34623828 | 125 |
| natriuretic peptide precursor type A | Nppa, Anp | Rn.PT.58.5865224 | 79 |
| natriuretic peptide precursor type B | Nppb, Bnp | Rn.PT.58.5595685 | 108 |
| tissue inhibitor of metalloproteases 1 | Timp1 | Rn.PT.58.34442920 | 127 |
| transient receptor potential cation channelC6 | Trpc6 | Rn.PT.58.18089975 | 94 |
| integrin beta 1 (fibronectin receptor beta) | Itgb1 | QT00187656 | 117 |
| connective tissue growth factor | Ctgf | QT00182021 | 102 |
| cyclophilin A | Ppia | Rn.PT.39a,22214830 | 140 |

**Table 2** **Characteristics of male sham-operated and AR animals at the end of the protocol.**

| Parameters | Sh (n = 6) | ShV (n = 6) | AR (n = 6) | ARV (n = 11) |
|---|---|---|---|---|
| Body weight, g | 586 ± 9 | 637 ± 31 | 599 ± 31 | 619 ± 12 |
| Tibial length, mm | 57 ± 0.2 | 57 ± 0.7 | 58 ± 0.3 | 58 ± 0.3 |
| Heart, mg | 1303 ± 30 | 1322 ± 68 | 1989 ± 54[*] | 1889 ± 87[*] |
| Heart/BW, mg/g | 2.2 ± 0.06 | 2.1 ± 0.06 | 3.3 ± 0.08[*] | 3.1 ± 0.11[*] |
| Left ventricle, mg | 1000 ± 28 | 976 ± 46 | 1538 ± 55[*] | 1426 ± 58[*] |
| Right ventricle, mg | 225 ± 9 | 247 ± 20 | 311 ± 28[*] | 325 ± 23[*] |
| Left atria, mg | 32 ± 2 | 30 ± 3 | 61 ± 3[*] | 58 ± 5[*] |
| Lungs, g | 2.6 ± 0.3 | 2.1 ± 0.2 | 2.5 ± 0.1 | 2.4 ± 0.1 |

Notes.

BW, body weight; M, males; F, females; V, valsartan.

Values are expressed as the mean ± SEM.

Group comparisons were made using two-way ANOVA followed by Holm-Sidak pos $t$-test for intergroup comparisons.

[*]$p < 0.05$ vs. respective sham group.

with valsartan. This again, was not observed in AR males. Lungs weight (a marker of overt heart failure) remained stable in the AR groups suggesting that the animals were still in the compensated state of the disease.

In Fig. 1, we illustrated individual variations in heart and heart chambers weights of AR animals relative to the mean of their respective sham-operated group in order to emphasize the extent of changes associated with the disease, the biological sex or the treatment. As expected, AR caused important cardiac hypertrophy in both male and female animals compared to sham. Valsartan had relatively little effects in blocking the development of

**Table 3  Characteristics of females AR animals at the end of the protocol.**

| Parameters | Sh (n = 5) | ShV (n = 6) | AR (n = 6) | ARV (n = 10) |
|---|---|---|---|---|
| Body weight, g | 318 ± 6 | 328 ± 3 | 337 ± 12 | 332 ± 9 |
| Tibial length, mm | 50 ± 0.4 | 50 ± 0.2 | 50 ± 0.5 | 50 ± 0.2 |
| Heart, mg | 836 ± 22 | 812 ± 26 | 1375 ± 107[*] | 1115 ± 59[***] |
| Heart/BW, mg/g | 2.6 ± 0.04 | 2.5 ± 0.07 | 4.1 ± 0.31[*] | 3.4 ± 0.14[***] |
| Left ventricle, mg | 638 ± 22 | 612 ± 24 | 1068 ± 81[*] | 862 ± 44[***] |
| Right ventricle, mg | 144 ± 3 | 149 ± 4 | 225 ± 20[*] | 171 ± 12[***] |
| Left atria, mg | 24 ± 1 | 15 ± 1[**] | 43 ± 4[*] | 32 ± 3[*] |
| Lungs, g | 2.1 ± 0.2 | 1.8 ± 0.3 | 2.0 ± 0.2 | 2.0 ± 0.1 |

**Notes.**

BW, body weight.

Values are expressed as the mean ± SEM.

Group comparisons were made using two-way ANOVA followed by Holm-Sidak pos $t$-test for intergroup comparisons.

[*] $p < 0.05$ vs. respective sham group.

[**] $p < 0.05$ vs. the respective untreated group.

cardiac hypertrophy in male animals. On the other hand, the hypertrophic response was slowed in female AR animals; this was significant for the left ventricle.

## Echocardiography data

As for the animal and heart characteristics described above, most echocardiographic parameters measured were significantly changed by AR (Tables 4 and 5). ARB treatment with valsartan of sham-operated rats had relatively no effects on echocardiographic parameters measured in this study (Table 4). Calculated ejection fraction, although still in the normal range was lower in treated sham males.

In AR animals, the effects of valsartan were observed on the LV wall thickness (intraventricular septal wall (SW) and posterior wall (PW), which were thinner in females compared to untreated AR rats. ARB had no effects in AR males (Table 5). Diastolic LV parameters were significantly altered in AR rats. Valsartan treatment had no effects on these parameters in males but significantly reduced E wave slope in females.

As illustrated in Fig. 2, LV dilation caused by AR was similar in rats of both sexes. However, LV septal and posterior walls thickening was more important in females. This was almost completely blocked by ARB treatment.

As illustrated in Fig. 3, both E wave and A wave were increased by AR in male and female rats. E wave slope, the most reliable marker of changes in diastolic function in our model was also increased (*Plante et al., 2003*). Valsartan treatment had no effects on these parameters in males but significantly reduced E wave and E wave slope in AR female.

## Markers of LV hypertrophy and extracellular matrix remodeling

We measured LV gene expression for several hypertrophy markers in AR animals relative to sham controls. *Anp* and *Bnp* mRNA levels were increased in both male and female AR animals (Fig. 4A). This increase was stronger for *Bnp* expression in females compared to males and was reversed by ARB treatment. Valsartan also reversed the increase in *Anp* expression in AR females. The expression of other hypertrophy markers was only changed

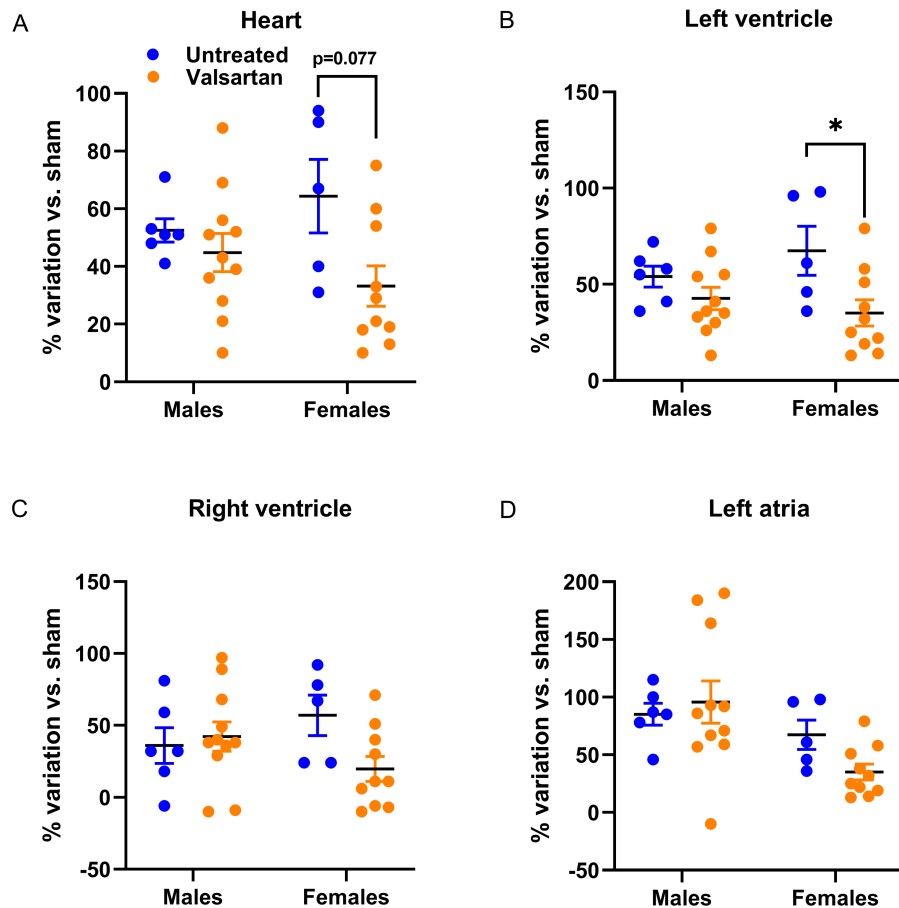

**Figure 1** **Effects of a 9-week treatment with valsartan on cardiac hypertrophy development caused by severe volume overload from AR.** Results are expressed as the percentage of variation of the indicated parameter compared to the mean of the same parameter for the respective sham-operated group. (A) Heart weight, (B) Left ventricular weight, (C) right ventricular weight and (D) left atrial weight. Results are expressed as the mean ± SEM. *: $p < 0.05$ between groups.

in AR males but not in females (Fig. 4B). Valsartan lowered *Fos* expression in both male and female animals but had little effects on the other two genes studied. *Trpc6* was up-regulated and *Klf15*, down-regulated in AR males. As expected, *Myh6* gene expression was reduced, by AR whereas *Myh7* expression increased. Valsartan treatment had no effect on the expression of both *Myh* genes.

Gene expression of extracellular components were only mildly regulated as illustrated in Fig. 5. *Col1* was significantly increased in female AR rats and this was reversed by ARB treatment. *Itgb1* encodes for beta-1 integrin, a sensor of mechanical stretch expressed on the surface of cardiac myocytes. In all AR groups with the exception of untreated males, *Itgb1* was up-regulated. *Timp1* was up-regulated in AR males and *Lox* in both males and females. The expression of the latter was reduced in ARB-treated males. *Ctgf* gene expression was increased in all AR groups but females treated with valsartan.

**Table 4  Echocardiographic parameters of male animals at the end of the protocol.**

| Parameters | Sh (n = 6) | ShV (n = 6) | AR (n = 5) | ARV (n = 6) |
|---|---|---|---|---|
| AR severity, % | NA | NA | 65 ± 4 | 64 ± 4 |
| EDD, mm | 7.7 ± 0.3 | 7.3 ± 0.3 | 10.7 ± 0.2* | 10.3 ± 0.3* |
| ESD, mm | 3.3 ± 0.1 | 3.6 ± 0.2 | 6.2 ± 0.1* | 6.0 ± 0.3* |
| SW, mm | 1.2 ± 0.06 | 1.1 ± 0.04 | 1.3 ± 0.03 | 1.1 ± 0.05 |
| PW, mm | 1.9 ± 0.02 | 2.1 ± 0.08 | 2.3 ± 0.02* | 2.4 ± 0.10 |
| RWT | 0.40 ± 0.032 | 0.44 ± 0.022 | 0.34 ± 0.013* | 0.35 ± 0.015* |
| EF, % | 81 ± 1 | 74 ± 2** | 66 ± 2* | 66 ± 2* |
| LV mass/BW, mg/g | 1.4 ± 0.16 | 1.3 ± 0.14 | 3.0 ± 0.15* | 2.7 ± 0.19* |
| HR, bpm | 393 ± 13 | 374 ± 10 | 365 ± 12 | 377 ± 11 |
| E wave, cm/s | 77 ± 3 | 74 ± 6 | 101* ± 9 | 99 ± 7* |
| A wave, cm/s | 38 ± 3 | 45 ± 6 | 57 ± 6* | 63 ± 5* |
| E wave slope | 2691 ± 52 | 2391 ± 115** | 3547 ± 407* | 3578 ± 180* |

**Notes.**

NA, non applicable; EDD, end-diastolic diameter; ESD, end-systolic diameter; SW, septum wall thickness; PW, posterior wall thickness; RWT, relative wall thickness; EF, ejection fraction; HR, heart rate; bpm, beats per minute.
Values are expressed as the mean ± EM.
Group comparisons were made using two-way ANOVA followed by Holm-Sidak pos *t*-test for intergroup comparisons.
*p < 0.05 vs. respective sham group.
**p < 0.05 vs. the respective untreated group.

**Table 5  Echocardiographic parameters of female animals at the end of the protocol.**

| Parameters | Sh (n = 6) | ShV (n = 11) | AR (n = 6) | ARV (n = 10) |
|---|---|---|---|---|
| AR severity, % | NA | NA | 68 ± 6 | 66 ± 3 |
| EDD, mm | 6.7 ± 0.2 | 6.5 ± 0.3 | 9.4 ± 0.2 | 8.7 ± 0.3 |
| ESD, mm | 2.9 ± 0.2 | 2.8 ± 0.3 | 5.3 ± 0.2 | 4.8 ± 0.3 |
| SW, mm | 1.1 ± 0.04 | 0.9 ± 0.07 | 1.3 ± 0.03 | 0.9 ± 0.05** |
| PW, mm | 1.7 ± 0.02 | 1.6 ± 0.10 | 2.7 ± 0.15* | 2.1 ± 0.07*** |
| RWT | 0.40 ± 0.032 | 0.44 ± 0.022 | 0.43 ± 0.013 | 0.38 ± 0.015*** |
| EF, % | 81 ± 3 | 81 ± 2 | 68 ± 3* | 70 ± 2* |
| LV mass/BW, mg/g | 1.9 ± 0.17 | 1.5 ± 0.11 | 5.0 ± 0.34* | 3.3 ± 0.16*** |
| HR, bpm | 413 ± 17 | 421+/- 14 | 386 ± 15 | 380 ± 9* |
| E wave, cm/s | 71 ± 4 | 86 ± 6 | 102 ± 6* | 94 ± 4 |
| A wave, cm/s | 40 ± 2 | 43 ± 3 | 65 ± 6* | 68 ± 4* |
| E wave slope | 2383 ± 129 | 2699 ± 127 | 4094 ± 251* | 3275 ± 190*** |

**Notes.**

NA, non applicable; EDD, end-diastolic diameter; ESD, end-systolic diameter; SW, septum wall thickness; PW, posterior wall thickness; RWT, relative wall thickness; EF, ejection fraction; HR, heart rate; bpm, beats per minute.
Values are expressed as the mean ± SEM. Group comparisons were made using two-way ANOVA followed by Holm-Sidak pos *t*-test for intergroup comparisons.
*p < 0.05 vs. respective sham group.
**p < 0.05 vs. the respective untreated group.

# DISCUSSION

In this study, we observed that angiotensin II receptor blockade using valsartan reduces the LV wall thickening taking place in AR female rats. We had previously shown that LV

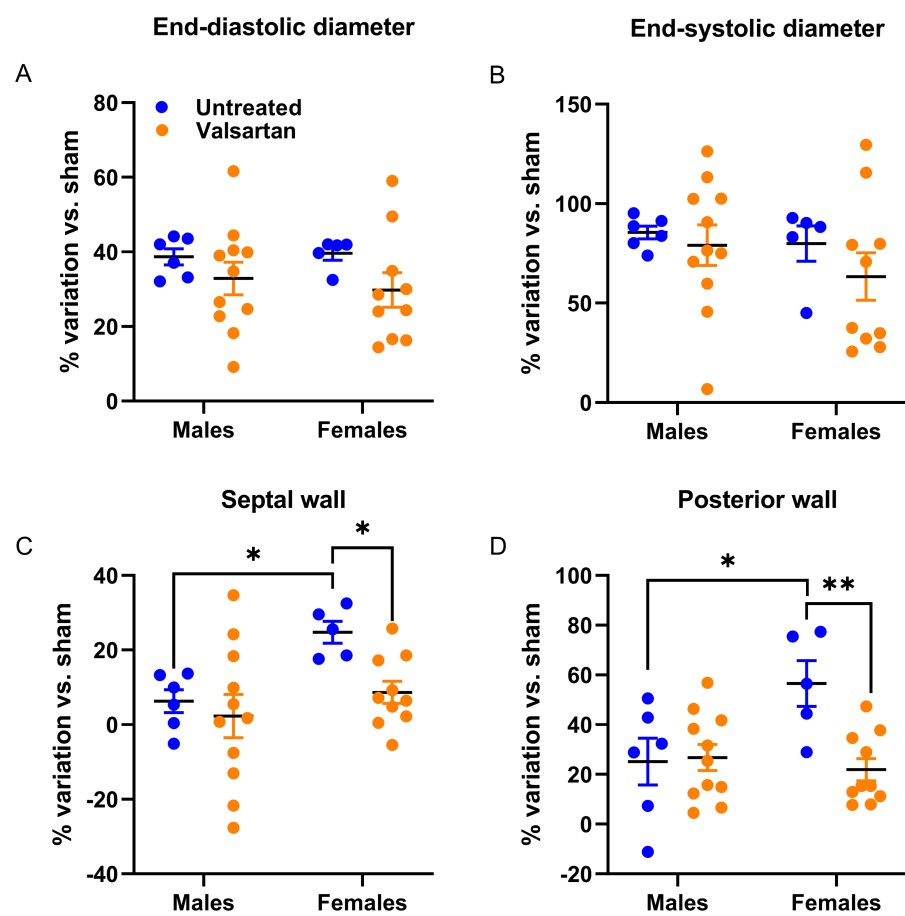

**Figure 2  Effects of a 9-week treatment with valsartan on LV hypertrophy development caused by AR on echocardiographic parameters.** Results are expressed as the percentage of variation of the indicated parameter compared to the mean of the same parameter of the respective untreated sham-operated group. (A) End-diastolic diameter, (B) end-systolic diameter, (C) inter-ventricular septal wall thickness and (D) posterior wall thickness. Results are expressed as the mean ± SEM. *: $p < 0.05$ and **: $p < 0.01$ between groups.

remodeling from AR in this model involves similar LV dilation in rats of both sexes but an excess of wall thickening in females. This results in somewhat maintained relative wall thickness (RWT), an index of LV remodeling (*Beaumont et al., 2017*).

The classic view of cardiac remodeling induced by a hemodynamic overload is that pressure overload (hypertension or aortic stenosis; afterload) is associated with initial concentric LV hypertrophy (wall thickening and equal or smaller chamber volume). On the other hand, volume overload (valve regurgitation; preload) induces chamber dilatation with no or little increase in wall thickness or eccentric remodeling (*Katz & Rolett, 2016*). This is probably more accurate for male animal models than for females as evidenced in the present study. We observed that eccentric LV remodeling took place in AR males resulting in a lower relative wall thickness ratio. In AR females, LV wall thickening concurrent to its dilation resulted in a maintained relative wall thickness ratio.

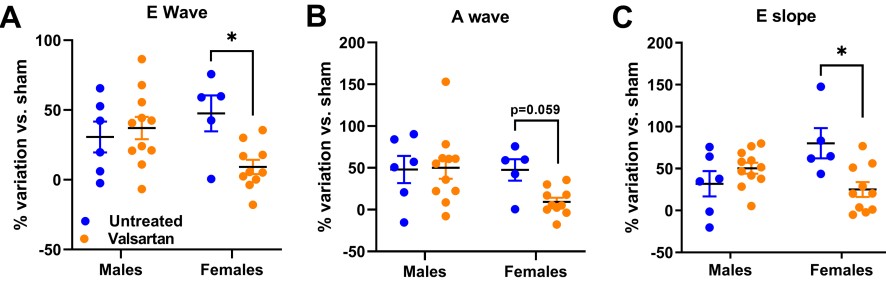

**Figure 3** **AR caused a general degradation of diastolic parameters that was improved by valsartan treatment in females but not in males.** (A) E wave, (B) A wave, (C) E wave slope. Results are expressed as the percentage of variation of the indicated parameter compared to the mean of the same parameter of the respective untreated sham-operated group. Results are expressed as the mean ± SEM. *: $p < 0.05$ between groups.

We observed that LV dilation was similar and relatively unaffected by angiotensin receptor blockade by valsartan both in males and females. Blood regurgitation from the aorta to the LV during diastole is a relatively stable determinant of the disease and cannot be modulated significantly by ARB. On the other hand, LV wall thickening, more important in female AR rats compared to males, was reversed by ARB leading to similar LV morphology between the sexes. Indexed LV mass estimated by echocardiography showed a similar trend where valsartan treatment partly reversed hypertrophy in females but not in males. The method we used makes the geometrical assumption of an elliptical LV. It is possible that the shape of the dilated LV in female AR rats does not obey this assumption.

Aortocaval fistula (ACF) is a model of global cardiac volume overload where sex differences have been studied with some details in the past in rats. Although this form of VO is less relevant for a clinical standpoint, it remains the most popular pre-clinical VO model in the literature. In 2002, Gardner and collaborators first reported that female rats with ACV developed less cardiac hypertrophy, evolved less towards heart failure and had better survival than males (*Gardner, Brower & Janicki, 2002*). Ovariectomy removed this advantage over males (*Brower, Gardner & Janicki, 2003*). A few years later, Dent and collaborators characterized this model further by echocardiography and at the molecular level. Both groups showed that estrogen could reverse the adverse effects of ovariectomy in females (*Dent, Tappia & Dhalla, 2010b*). In the AR rat model, we did not observe major effects related to the loss of estrogens by ovariectomy in females (*Drolet et al., 2006*).

Pharmacological interventions for cardiovascular diseases and heart failure are often less prescribed in women. Their absorption, distribution , metabolism and clearance is often different (*Humphries et al., 2017*). It is not excluded that the differences we observed here may have been related to sex-specific handling of ARB by females compared to males. Unfortunately, sex differences in the responseresponse to treatment in pre-clinical models of cardiovascular diseases have received little attention. In the pressure overload SHR (spontaneously hypertensive rats) model of LV hypertrophy, head-to-head comparison of treatment in animals of both sexes have seldom been performed. In 1982, Pfeffer and collaborators showed similar effects between male and female SHR of two anti-hypertensive
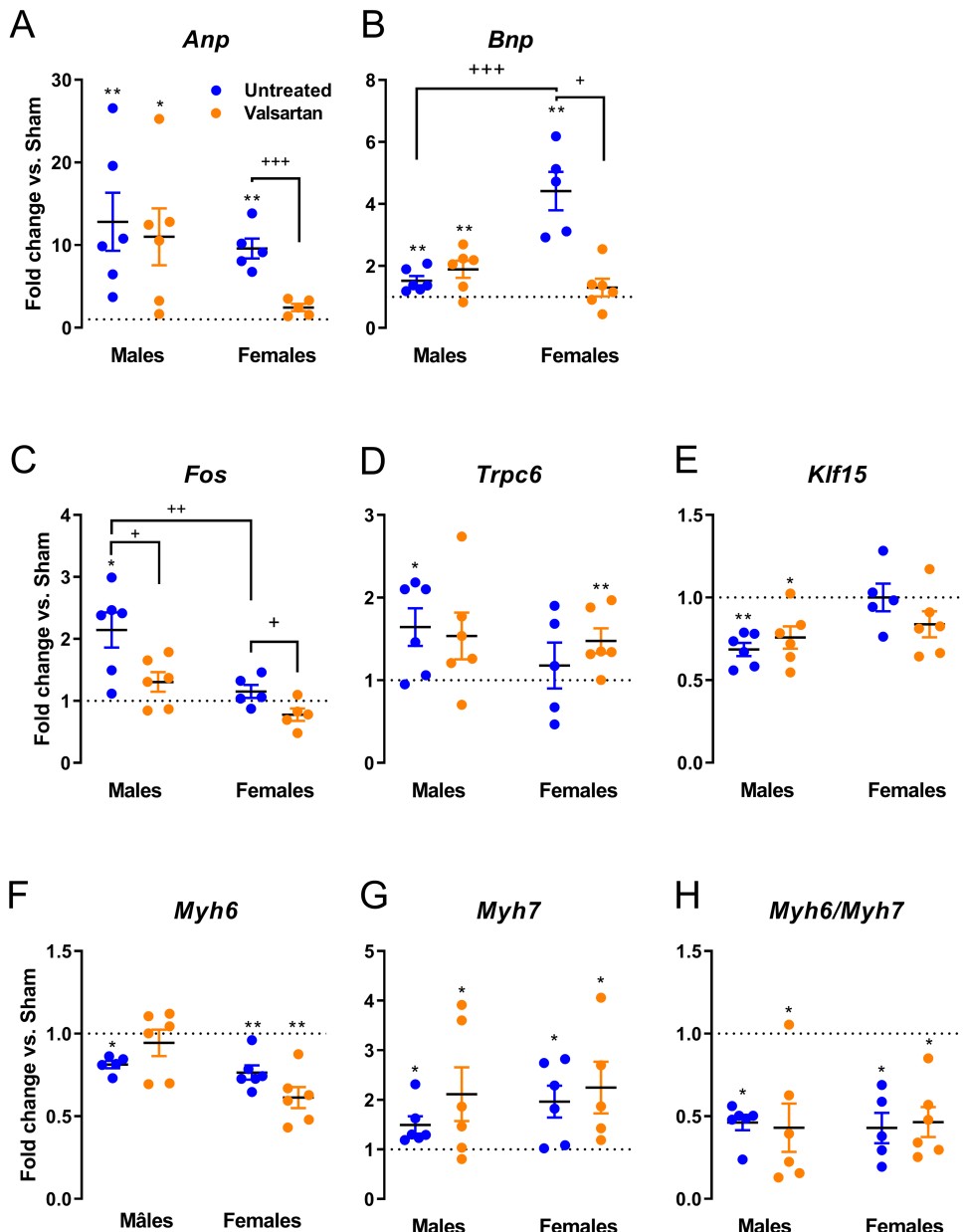

**Figure 4** **Evaluation by real-time quantitative RT-PCR of the LV mRNA leves of genes encoding for several hypertrophy markers in AR animals relative to sham controls.** (A) Anp, (B) Bnp, (C) c-Fos, (D) Trpc6, (E) Klf15, (F) Myh6, (G) Myh7 and (H) Myh6/Myh7 ratio. The results are reported in arbitrary units (AU) as the mean $\pm$ SEM ($n = 5 - 6$/gr.). Messenger RNA levels of the respective sham group were normalized to 1 and is represented by the dotted line. *: $p < 0.05$ and **: $p < 0.01$ vs. respective untreated sham group. +: $p < 0.05$, ++: $p < 0.01$ and +++: $p < 0.001$ between the indicated groups.

agents hydralazine and guanethidine on LV hypertrophy (*Pfeffer et al., 1982a*). Captopril (ACE inhibitor) has been shown to be effective to block LV hypertrophy in both males and females but was not compared in the same study (*Pfeffer et al., 1982b*; *Pfeffer et al., 1983*). More recently, the effects of vasopeptidase inhibitor omapatrilat and the ARB irbesartan

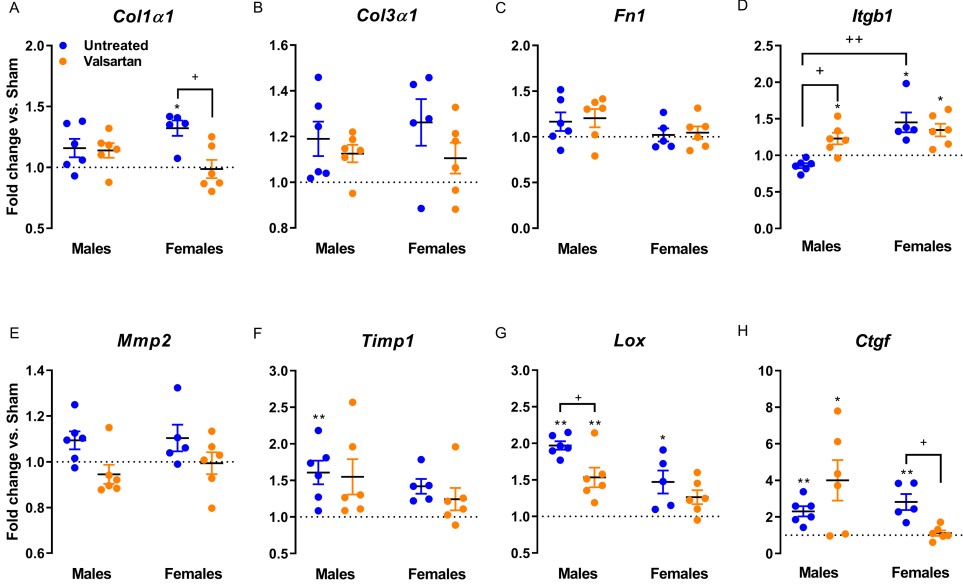

**Figure 5** **Evaluation by real-time quantitative RT-PCR of the LV mRNA levels of genes encoding for several extracellular matrix markers in AR animals relative to sham controls.** (A) Col1a1, (B) Col3a1, (C) Fn1, (D) Itgb1, (E) Mmp2, (F) Timp1, (G) Lox and (H) Ctgf. The results are reported in arbitrary units (AU) as the mean $\pm$ SEM ($n = 5 - 6$/gr.). Messenger RNA levels of the respective sham group were normalized to 1 and is represented by the dotted line. *: $p < 0.05$ and **: $p < 0.01$ vs. respective untreated sham group. +: $p < 0.05$ between the indicated groups.

in combination with a diuretic were studied in SHR/stroke prone male and female animals (*Graham et al., 2004*). Both regimen were efficient to lower LV hypertrophy development and this was similar for males and females. Romero and collaborators observed a better response of male SHR to atrial natriuretic treatment than for females although benefits were present for all animals (*Romero et al., 2015*).

Sex dimorphism in the response to treatment has not been studied before in VO rodent models. The present study design is also different from previous studies we made on male animals using this model (*Arsenault et al., 2013*; *Plante et al., 2004b*; *Zendaoui et al., 2011*; *Plante et al., 2009*; *Plante et al., 2008*). Here, we started treatment before the surgical induction of valve regurgitation instead of two weeks after and thus, when cardiac hypertrophy is already present. We choose to initiate treatment before AR induction in order to investigate the early implication of angiotensin II in the LV remodeling caused by severe VO instead of attempting to reproduce a more clinical situation. This study was also relatively short (2 months) instead of being more chronic (6 months and more) concentrating in early events. We consider that at the end of the protocol the animals were still in the compensated phase of the disease although both systolic and diastolic functions indicators were already significantly but not severely, altered. We showed in previous chronic studies that renin-angiotensin-aldosterone system (RAAS) inhibition using either angiotensin converting enzyme inhibitors such as captopril or angiotensin II receptor antagonists such as losartan can reduce the extent in LV hypertrophy, dilation and improve

survival in males (*Plante et al., 2009*; *Arsenault et al., 2013*). By comparing these studies performed in males to the present one, we noticed that the RAAS implication during the early stages of LV remodeling after AR induction does not seem to be as important as the one of the mTORC signaling pathway we observed in a previous study (*Drolet et al., 2015*). Rapamycin inhibition of mTOR signaling was able to reduced the extent of LV dilation in AR males, which was not the case here. It is possible that a higher dosage of valsartan may have provide a better inhibition. On the other hand, the dosage of valsartan we used in this study was similar to other studies performed in the past in rats ranging from 10 to 30 mg/kg/daily (*Li et al., 2002*; *Der Sarkissian et al., 2003*; *Tachikawa et al., 2003*; *Michel et al., 2016*).

The present study also enlightened that development of LV hypertrophy in this model first involves rapid LV dilation to accommodate the increased blood volume during diastole in males. In previous studies in males, we observed either a mild raise in systolic blood pressure or no changes (*Plante et al., 2009*; *Bouchard-Thomassin et al., 2011*; *Plante et al., 2004a*). Then, later in the disease, LV dilation and mass continue to increase and this can be blocked by inhibiting the RAAS (*Plante et al., 2004a*; *Arsenault et al., 2013*). Here, we showed that RAAS activation leads to LV wall thickening early in the disease. This helps maintain an enlarged but relatively normal LV morphology (relative wall thickness). Gain of LV mass is relatively as important as for AR males. In AR males however, our results seems to indicate that the RAAS blockade early in the disease is less consequential than later in the disease. LV hypertrophy development is rapid in our model during the first months and then slows later but still goes on (*Plante et al., 2003*). We can assume the first phase of LV remodeling is focused on the adaptation to pump the additional regurgitating blood. Then later in the disease, LV dilation continues and this can be blocked by either inhibiting angiotensin converting enzyme with captopril or ARB (*Plante et al., 2004b*; *Arsenault et al., 2013*). This inhibition of the RAAS later in the disease provides benefits such as less LV hypertrophy, better myocardial energy metabolism and better survival (*Arsenault et al., 2013*).

Expression of various LV genes associated with hypertrophy or extracellular matrix remodeling was assessed. In the case of *Anp* and *Bnp* genes, expression was more elevated in AR rats and valsartan normalized their expression but only in females suggesting that LV wall tension may have been improved by treatment. For three other hypertrophy markers, namely *Fos*, *Trpc6* and *Klf15*, gene expression was only altered in untreated AR males but not in females as previously reported (*Beaumont et al., 2017*). Interestingly, *fos* expression was lowered by ARB both in males and females. As for genes related to extracellular matrix components and metabolism, very few differences between the sexes or by treatment were registered. Volume overload is not associated with important myocardial fibrosis at least in the early steps of the disease (*Ryan et al., 2007*). This is also true in the AR rat model where collagen total myocardial content, at least in males, is still normal up to 9 months (*Lachance et al., 2009*).

It is not clear if the effects of angiotensin receptor blockade we observed in AR females provide benefits in the context of a LV VO. On one side, valsartan recreates in AR females an eccentric LV morphology similar to males, which tend to have a worse outcome

than untreated AR females after 6 months (*Beaumont et al., 2017*). On the other hand, it is probably that valsartan treatment may provide benefits as illustrated by several observations (*Beaumont et al., 2019*). First, diastolic function seemed to be improved by ARB as evidenced by less left atrial hypertrophy and better echocardiographic diastolic parameters. Valsartan helped reduced natriuretic peptides gene expression in females as well as collagen 1 and *Ctgf*. In males, benefits of valsartan were only observed for the decrease of *Fox* and *Lox* gene expression and an increase for *Itgb1*. Beta-1 integrin is one subunit of the dimeric fibronectin surface receptor. Its signaling promotes cardiac hypertrophy but also cardiac myocytes survival via Erk and Akt signaling pathways (*Brancaccio et al., 2006*).

We chose to not directly address the influence of sex hormones by castration or ovariectomy in this study. This would have added a level of complexity. We recently observed that loss of testosterone reduced LV hypertrophy in AR males and helped normalize the myocardial transcriptional profile suggesting an important role for this sex hormone in the response of the heart to a pathological stress *Beaumont et al., (2019)*. Additional studies are needed to better understand sex differences in the response to treatment in both pressure overload and volume overload pre-clinical models of LV hypertrophy. It is not clear how we can translate the observations made here and future ones to the human situation. On the other hand, our state of knowledge about heart diseases in women and how to treat them is still lagging (*Regitz-Zagrosek et al., 2010*; *Blenck et al., 2016*). We need even more basic knowledge to address this gap. We want to point out several limitations in this study. One is the relatively short duration of the protocol and as mentioned, the fact that we did not study the effects of sex hormones on the response to treatment. Aortic regurgitation is a relative rare disease in the Western world and is more prevalent in poorer countries (*Zühlke et al., 2017*). Since this disease is still lacking proven pharmaceutical options that could delay valve replacement if available for the patient, we consider that an effort on this is important.

In conclusion, we showed that female AR rats have a stronger early response to treatment with an angiotensin receptor antagonist, valsartan than males. This response is mainly concentrated on a female-specific feature of the LV remodeling in response to volume overload, LV wall thickening.

### Funding
This work was supported by operating grants from the Canadian Institutes of Health Research (MOP-106479) and the IUCPQ Foundation. The funders had no role in study design, data collection and analysis, decision to publish, or preparation of the manuscript.

### Grant Disclosures
The following grant information was disclosed by the authors:
Canadian Institutes of Health Research: MOP-106479.
IUCPQ Foundation.

## Competing Interests

The authors declare there are no competing interests.

## Author Contributions

- Élisabeth Walsh-Wilkinson performed the experiments, analyzed the data, prepared figures and/or tables, authored or reviewed drafts of the paper.
- Marie-Claude Drolet performed the experiments.
- Charlie Le Houillier and Ève-Marie Roy performed the experiments, analyzed the data.
- Marie Arsenault conceived and designed the experiments, prepared figures and/or tables, authored or reviewed drafts of the paper, approved the final draft.
- Jacques Couet conceived and designed the experiments, analyzed the data, prepared figures and/or tables, authored or reviewed drafts of the paper, approved the final draft.

## Animal Ethics

The following information was supplied relating to ethical approvals (i.e., approving body and any reference numbers):

The protocol was approved by the Universite Laval's Animal Protection Committee and followed the recommendations of the Canadian Council on Laboratory Animal Care.

## Data Availability

Animal characteristics and echocardiographic data are available in a Supplemental File.

## Supplemental Information

Supplemental information for this article can be found online at http://dx.doi.org/10.7717/peerj.7461#supplemental-information.

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
