# Peer review of "Sex differences in the response to angiotensin II receptor blockade in a rat model of eccentric cardiac hypertrophy"

_PeerJ, doi:10.7717/peerj.7461_

## Round 0.1 · original submission · Major Revisions

Dear Dr Couet,

We thank you for submitting your interesting study to PeerJ. The reviewers brought up some minor questions including a request that you supply direct comparison of sham animals and overload (AR) rats in figures and tables, compare heart to body weight data, and state the age of the rats used in the study, and rearrange data in line with suggestions of reviewer 2.

Clarification on the statistical methods used to analyze normality is also requested.

Reviewer 1 ·

Basic reporting

The manuscript is written in clear unambiguous professional English language. The references used are relevant and adequate and the authors provide enough background to show context.

Raw data are shared. However, the reporting of the gene expression could be improved. The data from the sham groups are missing. Additionally, the identification of the animals is also missing and the units of the data are missing in the table.

Experimental design

Walsh-Wilkinson et al study sex dimorphism and differential response to chronic treatment with valsartan in male and female rats with cardiac volume overload due to aortic valve regurgitation. Sex dimorphism in response to treatment in rodent models of volume overload has not been previously studied. The authors show that females show left ventricular wall thickening and that this effected is at least partially prevented by treatment with valsartan.

In general terms the study is well designed, and the results and analysis are correct. However, some aspects should be considered for improvement.

It would be helpful to present together, in the same tables and figures, the results in the sham group and in the aortic regurgitation (AR) groups and to include the statistical comparison between the sham untreated group and the AR untreated and treated groups. This would allow to see the effect of AR and to what extent valsartan prevented that effect. In its current for it is difficult to compare AR with Sham. Maybe males and females can be separated in different tables.

On the other hand, all the analysis related to cardiac weight are focused on raw cardiac and left ventricle weights. It would be interesting to index the weight either by body weight or by tibia length. Additionally, it would be useful to include the echocardiographically calculated left ventricular mass index.

From a methodological point of view, regarding mRNA analysis it is not stated which house-keeping gene was used for normalization.

A statement on the ethical approval should be included in the Methods section.

Minor comments:
- Lung weight data are not presented in the results.
- LOX is not included in the description of methods.

Validity of the findings

Regarding the statistical analysis, All the the analyses has been performed assuming a parametric distribution of the data but according to the methods this has not been checked. If possible data should be presented as dot plots instead of bar graphs.

As the authors state in the introduction they have previously shown in male rats that blockade of the renin-angiotensin system (Arsenault et al. 2013; Plante et al. 2009), had a beneficial effect in a similar experimental procedure reducing left ventricular hypertrophy and dilation. The lack of beneficial effect with valsartan in comparison to those previous studies should be discussed in depth. Why did valsartan have no effect on left ventricular dilation neither in males or females or on hypertrophy in males?

Overall, the conclusion is supported by the results.

As a minor issue, in the discussion an effect on systolic blood pressure is mentioned, but these results are not presented in the current study.

·

Basic reporting

- The abstract lacks contextual information. Please adapt in order for the abstract to include introductory information/background information in the beginning to place it into context and the perspective/importance/filling gap at the end. On the other hand, the results section in the abstract is too detailed and long.
- The reference format in the text may not be correct and the way in which references are presented now makes reading the text difficult. Please place references between brackets in the sentences.
- The start of the introduction lacks some flow, and it is difficult to understand the connections between the sentences. (lines 32-34)
- The introduction is very short, which makes it difficult to follow the argumentation for the study. Please add more background information on for example cardiac hypertrophy, volume overload and the RAAS system. Also,.
- The rationale and hypothesis of the study are unclear, and therefore the study design is difficult to assess. Why did you choose for a short study of 2 months instead of a chronic study? And why do you start treatment 1 week before surgery/AR induction, because this will not resemble a clinical situation (as medication will not be given before the problem arises)?

Experimental design

- The age of the rats is not stated in the methods section. From the weights in table 2&3 and the growth chart from Charles River, it seems that the female rats are younger than the male rats. If this is the case, female and male results cannot be directly compared, so then it will be better to show the results in different tables and discuss them separately in the context of responses to AR.
- What is the difference between the ‘complete echo exam’ (line 71-73) that is performed before AR induction and at the end of the protocol, and the ‘echocardiographic exam’ (line 75-76) that is performed 2 weeks after surgery and at the end of the protocol?
- In the methods, please add RNA isolation details.
- In the statistical analysis it is stated that student’s t-test is used when only 2 groups are compared (line 85). In the results, it is unclear when only 2 groups are compared and this test was used. How do you correct for multiple testing?

Validity of the findings

- Figure and table locations are not logic. Table 2&3 can be combined and 4&5 as well, because now it is difficult to compare the values of sham and AR rats. Figure 3 and 4 have to be placed in the text instead of at the end.
- Lung weight is an important factor to add to table 2/3, as it helps to assess whether rats are decompensating/have heart failure. Also, the baseline body weights will be informative, and if they changed due to surgery.
- The ShMV group has a high SEM for most measurements. Was the group normally distributed or were there any outliers?
- Please rewrite line 96-100 in a logical order taking into account the text before and the text from Figure 1.
- In Figure 1, there seem to be differences for some conditions but they are almost all not significant. It will be insightful to present the results in a scatter plot including the mean/median and error bars, so that the distribution of the samples can be seen.
- The legends to Figure 3 contain no information about subfigures A, B and C, so this has to be added. Also, please add ‘vs sham’ to the title of the y-axis behind fold change in figures 3&4, as this was also added in the other figures.
- Line 124 states that KLF15 is ‘upregulated’ by LVH in males, but figure 3B shows this must be ‘downregulated’.
- Line 126-127 seems contradictory to 127-129, because first you state that there are no important modulations and then significant changes are described. Please reformulate and replace ‘extracellular markers studied’ by mentioning the specific markers.
- Why did you choose to study the specific extracellular matrix genes? In the discussion, you state that extracellular matrix remodelling is low in the early phase of disease (line 211-213), so it is expected that there are no differences. Maybe it would be better to test for early matrix response genes like Thrombospondin-2, Intergrin β1 or CTGF, because collagen responses are later in the disease? In addition, other gene expression markers of cardiac remodelling exist that better reflect the earlier remodelling process, including markers for cardiomyocyte metabolism (e.g. angiopoietin-like 4) and for inflammation (e.g. Interleukin-6, TNF alpha).
- Line 162 misses a reference; to which side is ADME changed? (would you expect more or less effect of valsartan in females?)

Minor points:
- Line 12: ‘The aim of the study’ instead of ‘aim of study’.
- Line 24: ‘normalized’ or ‘helped normalizing’ instead of ‘help normalizing’.
- Line 37: HF has to be between brackets.
- Line 39: LV has to be written full, because this is the first time you use it in the main text.
- Line 43: ‘increase in’ instead of ‘increase’.
- Line 49: ‘as observed in…’ instead of ‘observed in …’, because it is a more general statement.
- Line 53: specify treatment; what kind of treatment?
- Line 81: why did you not use all animals, maybe better for significance/power?
- Line 96: ‘every parameter’ or ‘all parameters’ instead of ‘every parameters’.
- Table 4 legends: PW abbreviation description is missing.
- Line 120-121: This sentence can be interpreted like the increased expression of Bnp was higher as Anp, which is not the case, so include ‘compared to male’.
- Line 134: two times ‘relatively/relative’ in 1 sentence, so replace one.
- Figure 1 caption: ‘LV hypertrophy’ has to be ‘hypertrophy’, because not only LV is measured.
- Line 148-149: rephrase sentence
- Line 164: rephrase sentence or leave ‘since’ out.
- Line 190-191: Is there a study that used a higher dose of Valsartan?

Additional comments

this is a nice, well-written and timely paper. Addressing the comments will improve it further.

---

## Round 0.2 · accepted · Accept

The revised manuscript has been significantly improved.